# RNAi-Mediated Silencing of *Laccase 2* in *Culex pipiens* Pupae via Dehydration and Soaking Results in Multiple Defects in Cuticular Development

**DOI:** 10.3390/insects15030193

**Published:** 2024-03-14

**Authors:** Anastasia N. Naumenko, Megan L. Fritz

**Affiliations:** Department of Entomology, University of Maryland, College Park, MD 20742, USA

**Keywords:** RNAi, *Culex pipiens*, *laccase 2*

## Abstract

**Simple Summary:**

While RNA soaking was successfully used in *Aedes* pupae, no such technique has yet been developed for *Culex* mosquitoes. Here, we offer a quick and easy assay for screening the functional role of genes in *Culex pipiens* mosquitoes without using microinjections. By dehydration of early stage pupae and subsequent rehydration in highly concentrated dsRNA, we were able to achieve knockdown of the *laccase 2* gene responsible for melanization and sclerotization of the mosquito cuticle. Our results suggest this assay will be useful for the functional screening of genes expressed in early pupal to early adult stages.

**Abstract:**

Mosquitoes transmit a range of pathogens, causing devastating effects on human health. Population genetic control strategies have been developed and successfully used for several mosquito species. The most important step in identifying potential targets for mosquito control is the understanding of gene function. RNA interference (RNAi) is a powerful tool for gene silencing which has been widely used to study gene function in insects via knockdown of expression. The success of RNAi in insects depends on the efficient delivery of dsRNA into the cells, with microinjections being the most commonly used to study mosquito gene function. However, microinjections in the pupal stage lead to significant mortality in *Aedes* and *Culex* species, and few studies have performed microinjections in Culicinae pupae. Advanced techniques, such as CRISPR/Cas9 knockout, require establishing individual mosquito lines for each gene studied, and maintaining such lines may be limited by the insect-rearing capacity of a laboratory. Moreover, at times gene knockout during early development (embryo stage) has a deleterious effect on mosquito development, precluding the analysis of gene function in the pupal and adult stages and its potential for mosquito control. There is a need for a simple procedure that can be used for the fast and reliable examination of adult gene function via RNAi knockdown. Here, we focus on the aquatic stages of the mosquito life cycle and suggest a quick and easy assay for screening the functional role of genes in *Culex pipiens* mosquitoes without using microinjections. By dehydration of early stage pupae and subsequent rehydration in highly concentrated dsRNA, we achieved a moderate knockdown of *laccase 2*, a gene that turns on in the pupal stage and is responsible for melanization and sclerotization of the adult cuticle.

## 1. Introduction

Mosquito-borne diseases continue to be a significant health concern, causing millions of deaths globally. Traditional broad-spectrum methods of vector control, such as insecticide spraying and bed nets, have proven to be effective but have limitations in terms of sustainability and the emergence of insecticide resistance. The discovery of RNA interference (RNAi)—a virus defense tool naturally found in eukaryotic organisms [1]—offered an opportunity for the development of species-specific approaches for mosquito control.

RNAi can be used to manipulate gene expression via post-translational gene silencing by degrading or inhibiting the translation of gene-specific messenger RNA (mRNA) molecules [2]. In mosquitoes, the RNAi pathway has been exploited for decades to study gene function, as well as for the development of RNAi-based vector and pathogen management approaches [3]. Methods of dsRNA delivery in mosquitoes are diverse and vary according to their intended application.

The development of field-relevant RNAi-based mosquito management strategies often exploits the natural feeding behavior of mosquitoes for dsRNA delivery. Mosquito larvae feed on microorganisms (e.g., bacteria, yeasts) in their aquatic habitats, and these can be genetically engineered to produce dsRNA that silences the expression of critical fitness-related genes. For example, in *Aedes aegypti*, two chitin synthase genes were successfully knocked down by adding bacterial lysate containing dsRNA into rearing water [4]. In another species, *Ae. albopictus*, a viral (densovirus)-mediated interference system was developed for larvae, downregulating V-ATPase synthesis and causing high mortality [5]. Microbial methods are also being developed for adult control. In *Anopheles gambiae*, dsRNA in bacterial lysate mixed with glucose solution was used as a sugar source for feeding adult mosquitoes [4,6]. Mysore and co-authors [7] have also employed genetically engineered yeast that produced short hairpin RNAs (shRNAs) to suppress *Rbfox1* genes in larvae and adult *Aedes albopictus*, *Anopheles gambiae*, and *Culex quinquefasciatus* via feeding, causing high larval and adult mortality.

Alternative nonmicrobial feeding-based delivery methods include feeding larval and adult stages with nanoparticles as dsRNA carriers. Nanoparticles can protect dsRNA from degradation by endogenic nucleases and improve its stability during digestion processes in the midgut. The potential of various nanoparticles has been explored, such as chitosan, carbon quantum dots, and silica nanoparticles [8,9,10,11], and Effectene (transfection reagent)-based liposomes [8]. While nanoparticles offer great potential in field applications, challenges remain in terms of optimizing nanoparticle formulations for maximum dsRNA loading, stability, and appropriate release within the mosquito body. The efficiency and specificity of dsRNA uptake by feeding varies according to life stage, and feeding-based dsRNA delivery is not feasible for the management of mosquitoes in embryonic or pupal (non-feeding) life stages.

To study gene function, the delivery of dsRNA into mosquitoes may occur by feeding but also by microinjection, electroporation, or soaking. Microinjection can be performed at virtually every life stage, starting from an embryo, and is the most common technique used for gene knockdown in mosquitoes [9,10,12]. Although effective, microinjection involves physically puncturing the mosquito cuticle with a fine needle and is a delicate procedure that requires considerable skill and precision. The procedure can cause localized tissue damage, which may affect both mortality rates and experimental outcomes [10,12,13]. High mortality rates have been recorded for microinjections of *Culex* pupae [12,13,14], for example. Electroporation utilizes an electrical pulse to facilitate dsRNA delivery. It was first successfully employed in Odonata larvae and later used to facilitate the knockdown of *AsCPR128* in *An. sinensis* [15]. Both methods may be challenging to scale up for high-throughput testing of gene function or screening for potential vector control targets, although microinjections have been used for this purpose [16].

Soaking is an alternative method of dsRNA delivery that involves immersing mosquito larvae or pupae in a solution containing dsRNA. The dsRNA can penetrate exposed cell membranes and reach the cytoplasm, thus initiating the RNAi pathway. Soaking assays are less technically challenging compared to microinjections because they do not require expensive equipment and can also target the non-feeding pupal stage. Double-stranded RNA delivery by soaking has been efficiently used in several studies. For example, soaking was first applied to first instar *Ae. aegypti* by Singh and co-authors. They suppressed the expression of *chitin synthase-1* and *-2*, *heat shock protein 83*, and *β-tubulin* genes by soaking larvae for 2 h in dsRNA [17]. This same delivery method was successfully used to knock down the expression of fertility genes in *Ae. aegypti* by soaking first through fourth instars [18]. Another study demonstrated the knockdown of the gene encoding *heat shock protein 90* in *Cx. pipiens* by dehydration of larvae in salt solution and subsequent rehydration in dsRNA [19].

Most of the above-listed approaches target genes expressed in the larval and adult stages. However, functional studies for genes initially expressed in the pupal stage and critical for adult development are limited because this life stage does not feed and is fragile. Recently, the *CYP4G35* gene, which is expressed at high levels in the pupal stage, was efficiently suppressed in *Ae. aegypti* pupae simply by soaking them in water with added dsRNA. This effect lasted into early adulthood [20]. To our knowledge, no studies have yet used soaking for dsRNA delivery to study gene function in *Culex* pupae. 

Here, we propose a rehydration-driven soaking assay that can be utilized for the efficient delivery of dsRNA in life stages and organisms where microinjection and feeding protocols are challenging or impractical. Soaking is an attractive approach to dsRNA delivery, particularly when screening large numbers of pupae or genes because it reduces the handling time of individual insects over microinjections. As proof of concept for our dsRNA delivery method, we targeted *laccase 2*, a gene encoding an enzyme that plays a crucial role in the melanin synthesis pathway, in *Cx. pipiens*. Laccase 2 is involved in the final steps of melanin synthesis and utilizes molecular oxygen to oxidize phenolic substrates. This oxidative reaction generates reactive quinones, which can then undergo polymerization to produce melanin [21]. The knockdown of *laccase 2* expression should disrupt melanin production and consequently impair crucial physiological functions in mosquitoes. Based on previous studies of *Tribolium castaneum* [22] and *An. sinensis* [23], *laccase 2* showed promise as a gene target for this silencing proof-of-concept study. Knockdown mutants with reduced *laccase 2* expression showed defects in melanization. Moreover, in *An. sinensis*, the pale cuticle was significantly thinner in *dsAsLac2* mosquitoes, and the immune response to pathogens was weakened. We reasoned that if *laccase 2* has a similar functional role in *Cx. pipiens*, the defects resulting from the knockdown of expression would allow for easy phenotypic scoring following dsRNA delivery via soaking. While *laccase 2* served as a straightforward gene target for our proof of concept, the manipulation of *laccase 2* expression can also aid in understanding its specific physiological role in *Cx. pipiens* and potential application to mosquito control.

*Cx. pipiens* pupae are able to survive short periods of desiccation. We leveraged this to silence *laccase 2* expression by the dehydration and subsequent soaking of newly molted pupae in concentrated dsRNA. The knockdown of *laccase 2* expression resulted in visible alterations in mosquito phenotype and fitness. The pupae in our assay had much lower mortality rates compared to other assays, perhaps because dehydration, soaking, and rearing were performed in the same compartment, reducing handling. Our work establishes pupal dehydration and soaking as a feasible approach for studying genes that are expressed and functional during the pupal stage. 

## 2. Materials and Methods

### 2.1. Mosquito Rearing

Mosquito colonies (*Cx. pipiens* aboveground Northfield and belowground Cal1 populations) were maintained at 25 °C and 60–70% humidity under a photoperiod of 16 h light–8 h dark cycle. Larvae were fed with bovine liver powder every 48 h and kept at low density to maximize the number of simultaneously pupating individuals. Further details on collection and rearing procedures for these strains can be found in Noreuil and Fritz [24]. 

### 2.2. Analysis of Laccase Protein Sequence and Structure 

To identify the *laccase 2* gene among 5 laccase orthologs in *Cx. quinquefasciatus*, all homologous *Culex* laccase sequences were retrieved from the NCBI database (http://www.ncbi.nlm.nih.gov/, accessed on 1 April 2022) and Vectorbase (http://www.vectorbase.org/, accessed on 1 April 2022) using the BLASTP program. They included CPIJ012357-PA, CPIJ010466-PA, CPIJ000864-PA, CPIJ000865-PA, and CPIJ000866-PA. We also obtained orthologous Laccase 2 sequences from other insects with the following accession numbers: *Anopheles gambiae* (Lac1: AGAP003738-PA; Lac2:AGAP006176-PB), *Anopheles atroparvus* (Lac1: AATR020818-PA; Lac2: AATR020769-PA), *Anopheles sinensis* (ASIC018958-PA), *Aedes aegypti* (Lac1: AAE026602; Lac2: AAE016992-PA), and *Tribolium castaneum* (Lac1: NP_001034514; Lac2: NP_001034487). Tyrosine hydroxylase amino acid sequences from *Cx. quinquefasciatus*, *Tr. castaneum*, and *An. sinensis* served as an outgroup (CPIJ014156-PA, NC_007418.3, and ASIC011233-PA, respectively). ClustalW (Kalign [25]) was used for multiple protein sequence alignments, and MEGA11 [26] was employed to construct a phylogenetic tree using the maximum likelihood (ML) analysis. The bootstrap values were obtained based on 1000 bootstrap replications. To predict protein structure and confirm the similarity of *Cx. quinquefasciatus* Laccase 2 to other Laccase 2 structures, we used the AlphaFold Protein Structure Database [27].

### 2.3. Semi-Quantitative PCR and Temporal Expression Analysis of CpLac2

The timing of *laccase 2* expression initiation in pupal and adult *Cx. pipiens* (*CpLac2*) was previously unknown. To establish an optimal time for pupal soaking, three pupae were collected at 0, 1, 2, 5, 18, 24, 28, 32, and 47 h after pupation (2 biological replicates) for temporal expression analysis of *CpLac2*. Total RNA was extracted from pooled mosquito samples with a QIAGEN RNeasy Mini kit (Germantown, MD, USA, cat. no. 74104) using the manufacturer’s instructions. Additionally, 1000 ng of the resulting RNA template was used in the cDNA synthesis reaction. cDNA was produced using a BIO-RAD iScript cDNA Synthesis Kit (Hercules, CA, USA, cat. no. 1708890) according to the manufacturer’s instructions. For the semi-quantitative polymerase chain reaction (PCR), cDNA was amplified using the standard protocol for HotStart ImmoMix Polymerase (BIOLINE USA Inc., Taunton, MA, USA, cat. no. BIO-25020). A negative control containing no template and *actin* primers was included. PCR was performed with a 10 min initial denaturation at 97 °C, followed by cycles of 10 s at 95 °C, 30 s at 58 °C, 30 s at 72 °C, and a final extension for 5 min at 72 °C. Our semi-quantitative approach included multiple rounds of PCR replicated across samples and with varying numbers of cycles from 25 to 35. The timing of *laccase 2* expression initiation was determined by the visualization of amplicons on a 2% agarose gel using *actin* primers as an internal control. Gel images showed the clearest differences in the initiation of expression after 35 PCR cycles, and subsequent analysis of the expression differences focused on samples amplified under those thermal cycler conditions. All primer sequences are listed in Table 1. 

The relative quantification of band intensities was performed using Image J (ImageJ bundled with 64-bit Java 8, v.1.8.0_345) gel analysis [28,29]. The *laccase 2* band intensities were standardized to *actin* band intensities for each sample loaded onto the gel. Fold change in band intensities were calculated relative to the *actin* band intensity at 24 h timepoint using the following formula: FC = (Y_n_/Y_max_)/(Z_n_/Z_max_),
where Y_max_ = *laccase 2* band intensity at 24 h and Y_n_ = *laccase 2* band intensity at a single time point. Z_max_ = *actin* band intensity at 24 h and Z_n_ = *actin* band intensity at the same time point selected for Y_n_.

### 2.4. Knockdown of CpLac2 and Test of Pupal dsRNA Soaking

*Generation of Linear DNA Template for dsRNA synthesis.* To generate templates for in vitro dsRNA synthesis, we used HotStart ImmoMix Polymerase mix (BIOLINE USA Inc., Taunton, MA, USA, cat. no. BIO-25020) and primers containing T7 promoters from the 5′ end (Table 1). The annealing temperature for the first 5 PCR cycles was 60 °C, followed by an additional 34 cycles performed at Tm 69 °C. Five μL of the product was examined on a 2% agarose gel before in vitro transcription (IVT) was performed to assess the product quality and concentration, as well as to verify that the products were unique (only one band visible) and of the expected size. PCR products were excised from the gel and purified using a Wizard SV Gel and PCR Clean-Up Kit (Promega, Madison, WI, USA, cat. no. A9281). Before using the PCR template in an IVT reaction, the product was cut from the gel, extracted using the Wizard SV Gel and PCR Clean-Up System (Promega, Fitchburg, WI, USA), and additional PCR was performed with the same primers to generate more template. Product concentrations were measured with a NanoDrop 1000 Spectrophotometer (Thermo Fisher Scientific, Waltham, MA, USA), and 10 μL of PCR product was sent for Sanger sequencing from the 5′ and 3′ ends. The resulting sequences were aligned with the target genomic sequence to validate the product specificity.

*dsRNA Preparation*. To design primers for dsRNA synthesis, we used Primer3 v4.1.0 software [30,31,32]. The primers are listed in Table 1. Three dsRNA constructs were employed in the assays: 437 bp *dsLaccase 2-5′* spanning exons 2 and 3 close to the 5′-end of the gene; 379 bp *dsLaccase 2-3′* spanning exons 6, 7, and 8 close to the 3′-end of the gene (Figure 1A); and *dsGFP* (used as a negative control), 699 bp amplified from 3788..4486 Turbo GFP locus of circular plasmid (Addgene, supplied by L. Pick). Double-stranded RNA was synthesized using an in vitro transcription MEGAscript T7 Transcription Kit (Thermo Fisher Scientific, Waltham, MA, USA, cat.no. AM1334). The reaction was incubated overnight at 37 °C. After the incubation, a brief DNA digestion with DNAse was performed to remove the remaining template. The whole reaction was then denatured in a heat block at 97 °C for 3 min and slowly cooled to room temperature to ensure the annealing of RNA strands. One μL of the product was then mixed with 3 μL nuclease-free water and examined on 2% agarose gel containing 1% bleach. The dsRNA product was then mixed with 3.5× volumes of ice-cold isopropanol and 0.1× volume of sodium acetate and precipitated overnight at −20 °C. The resulting dsRNA pellet was dissolved in 80–100 μL of nuclease-free water, and dsRNA was stored at −20 °C. The product concentration was measured using a NanoDrop 1000 Spectrophotometer (Thermo Fisher Scientific, Waltham, MA, USA).

*Dehydration and soaking in dsRNA*. All pupae were removed from larval rearing pans 2 h before assay setup. In the subsequent 2 h window, freshly molted pupae that were pale in color were removed from the rearing pans, washed in cups containing deionized water, and placed in individual cells of 96-well cell culture plates using a trimmed plastic Pasteur pipette. Water that was inadvertently added was removed, and the well was dried using Kimwipes. Pupae were desiccated at room temperature for 30 min. After 30 min, 10 µL of the solution containing 1 μg of either *dsLaccase 2-5′* (hereafter *dsLac2-5*), *dsLaccase 2-3′* (hereafter *dsLac2-3*), *dsGFP* (negative control), or ddH_2_O mixed with Brilliant Blue dye (0.001 mg/mL) was added to each well. The mortality did not differ significantly between the *dsGFP* and ddH_2_O groups; therefore, the ddH_2_O group was excluded from further testing after our first replicate. After another 30 min, any dead individuals were removed, dye uptake was validated using a binocular scope (Olympus SZ61 stereoscope, Olympus America Inc., Waltham, MA, USA), and 150 μL of deionized water was added to each well containing the pupae. The mortality in experimental and control groups was assessed at 1 h, 24 h, and 47 h timepoints following desiccation. Cell culture plates containing pupae were held in mesh cages. After mosquitoes started to emerge, the cages were monitored every 30 min until the last mosquito emerged. The phenotypes of adults in the control and experimental groups were evaluated at 1 h post-emergence. To examine gene expression, three pupae from each timepoint were saved in 500 μL of RNAlater™ RNA-stabilizing solution (Thermo Fisher Scientific, Waltham, MA, USA AM7021) and held at 4 °C until RNA isolation. Total RNA was extracted from the pooled mosquito samples and cDNA was synthesized as described above for 3 pupae at 1, 24, and 47 h after the soaking assay for each of the replicates. The remaining pupae were allowed to emerge as adults and their phenotypes were scored. Per treatment, there were 20–36 pupae in each replicate, and the experiment was performed in 6 biological replicates. 

*dsRNA Microinjections*. To further confirm that *dsLac2* knockdown phenotypes were associated with dsRNA and not desiccation, we performed dsRNA microinjections. Twenty 2–3 h-old pupae were collected and rinsed in a Petri dish filled with deionized water. Individual pupae were transferred onto a 60-well cell culture microplate using a trimmed plastic Pasteur pipette. Microinjection needles were pulled using a Sutter P-97 micropipette puller (Sutter Instrument Company, Novato, CA, USA). After loading the dsRNA solution, the microinjection needle was inserted into the tip of a Pasteur pipette and fixed with parafilm. The injection needle was inserted by applying light pressure and sliding movements to the area near the 5–6th antennal segments under the antennal plate. Approximately 100 nL of 2 µg/µL *dsLac2-5* (or *dsGFP* for the control) solution mixed with Brilliant Blue dye (0.001 mg/mL) was injected into the hemolymph according to gradations added to the needle. Pupae were gently transferred into a new cup containing 5 mL of ddH_2_O treated with 50 µL of 100× penicillin/streptomycin solution (10,000 UI/mL penicillin/10,000 µg/mL streptomycin, Millipore Sigma, Milwaukee, WI, USA) and placed into mesh mosquito cages. After adults started to emerge, treatment and control groups were monitored every 30 min until the last mosquito emerged. Adult mosquitoes were assessed at 1 h post emergence. Dead pupae were removed and counted at 1 h, 24 h, and 47 h after the assay, and samples for RNA extraction were taken from the *dsLac2-5* and control groups at 24 and 47 h. 

### 2.5. qRT-PCR Evaluation of Gene Expression

*Laccase 2* expression in the knockdown and control sample pools was determined by qRT-PCR with *actin* primers as an internal control. One µg of DNase-treated RNA was used for cDNA synthesis. cDNA was diluted 10× in nuclease-free water and used as a template in the qRT-PCR assay. In a 10 µL qRT-PCR, 0.5 µL of cDNA was used. Each sample was run in duplicate wells of a 96-well plate. No template and no RT controls were included in each reaction. qRT-PCR was performed on a Roche Lightcycler 480 System [33] using SYBR Green master mix (Bio-Rad Laboratories, Inc. Hercules, CA, USA) and *laccase 2*-specific primers (Table 1). The reactions were performed for 1 min at 95 °C, followed by 39 cycles of 10 s at 95 °C, 30 s at 58 °C, and 30 s at 72 °C. The melting curve was analyzed at 60–95 °C to ensure a single peak, and thereby primer specificity. An initial standard curve analysis was performed for both *laccase 2* (target) and *actin* (internal control) genes (primers are listed in Table 1). Primer efficiencies were calculated as (10^(−1/The Slope Value)−1) * 100 and were 93.8% and 95.5% for *laccase 2* and *actin*, respectively.

### 2.6. Phenotypic Evaluation

Mosquito phenotypes were evaluated via an Olympus SZ61 stereoscope at 30X magnification. Deformations to the cuticle, wings, legs, and body segments, as well as a lack of pigmentation, were recorded. Inability to successfully emerge from the pupal enclosure, as was observed by Du and co-authors [23], was also documented. Mosquitoes were scored as having a knockdown phenotype if we observed one or more of the traits described above. 

### 2.7. Data Analysis

A statistical analysis of mosquito survivorship following treatment with dsRNA was performed using a Mantel–Cox log-rank test for survival curves with GraphPad Prism (version 10.0.0 for Windows, GraphPad Software, Boston, MA, USA). The differential expression of *laccase 2* was quantified in pupae exposed to *dsLac2-5*, *dsLac2-3*, and *dsGFP* at multiple timepoints following exposure with the ΔΔCt method according to [34]. The phenotypes produced by *dsLac2-5* and *dsLac2-3* were similar, as were the reductions in gene expression following dsRNA treatment. Therefore, the samples exposed to both dsRNAs targeting *laccase 2* were analyzed together in comparison to dsGFP-exposed pupae collected at the same timepoint and run together on the same qRT-PCR plate. Welch’s unpaired t-tests of the ΔCt values were used to test for statistically significant gene expression differences with an a priori α-value of 0.05. 

## 3. Results

### 3.1. Phylogenetic Analysis of Cx. pipiens Laccase 2

The results of our phylogenetic analysis demonstrated that *CPIJ010466* was the *Cx. quinquefasciatus laccase 2* gene. In general, protein sequences used to build our phylogenetic tree were clustered into two distinct clades, Laccase 1 and Laccase 2. CPIJ010466 was grouped into the Laccase 2 group and CPIJ012357 was grouped into the Laccase 1 group (Figure 1B). Three other laccase-like proteins (CPIJ000864, CPIJ000865, CPIJ000866) with unknown functions were clustered together in a separate group. Further investigation of the gene structure indicated that it consisted of nine exons encoding a 2196 bp transcript (Figure 1A), with a protein sequence length of 731 AA. Like other insect laccases, the predicted structure of the enzyme encoded by *laccase 2* in *Cx. quinquefasciatus* consisted of six regions: a signal peptide, a non-conserved amino-terminal region, a rich cysteine-rich region, and three Cu-oxidase domains (I, II, and III) [35] (Figure 1C).

### 3.2. Initiation of Cuticle Darkening Coincides with Laccase 2 Peak Expression

*Cx. pipiens* pupae normally require around 48 h until adult emergence. Freshly molted pupae were pale and translucent in color (Figure 2A(a)), and by 1–2 h post-pupation, some darkening of the dorsal segments occurred. However, the body color remained light until 18–24 h into the pupal stage (Figure 2A(a–c)). At approximately 24 h, significant darkening of the dorsal segments, antennae, eyes, and legs was initiated (Figure 2A(d,e)). By 47 h post-pupation, the body and legs became dark (almost black) and the adult mosquito was ready to emerge (Figure 2A(f)).

A comparison of *laccase 2* expression patterns with this time course of melanization in *Cx. pipiens* pupae showed that expression levels were at their lowest immediately after pupation (Figure 2B, Table 2). As pupal development progressed, the expression of *laccase 2* increased (Table 2). At around 18 h post-pupation, *laccase 2* expression was significantly upregulated and remained high until 48 h. Figure 2B, Table 2). Peak expression of *laccase 2* occurred between 18–24 h and coincided with the start of darkening in the adult cuticle during metamorphosis, supporting a role for *laccase 2* in the melanin synthesis pathway of *Cx. pipiens*. After the 28 h timepoint, *laccase 2* expression subtly decreased (Table 2). 

### 3.3. Dehydration and Soaking with dsRNA Suppresses Pupal Expression of Laccase 2

A qRT-PCR assay was used to confirm the knockdown of *laccase 2* expression and revealed that dehydration followed by soaking in *dsLac2* reduced expression by approximately half of the control treatment levels at the 1 h and 24 h pupal collection timepoints (Table 3). Differential gene expression levels were statistically significant at the 24 h timepoint. Expression of *laccase 2* continued to be suppressed at the 47 h timepoint but rose to 69% of the control levels; due to the variability among replicates, this was not statistically significant. We also examined differences in *laccase 2* expression among pupae injected with *dsLac2-5* and *dsGFP*. In the single *dsLac2-5* injected replicate, *laccase 2* expression dropped to ca. 18–20% of the control levels at both the 24 h and 48 h collection timepoints. 

### 3.4. Laccase 2 Suppression Affects Mosquito Population Fitness, Causing a Broad Range of Defects

Initial assessment of the impact of *laccase 2* suppression on pupal mortality revealed rates of 29.4–19.17% and 12.5% for the target and control groups, respectively (Figure 3A,B). These rates were lower than those described by Arshad and co-authors [20] in their assay (56.92% and 56.92%) of dsRNA soaked (treatment and control) *Ae. aegypti* pupae. Among our three treatments, the *dsLac2-5* group experienced the lowest rates of survivorship at 70.6 ± 3.2% versus 80.83 ± 2.8% and 87.5 ± 2.3% for *dsLac2-3* and *dsGFP*. This difference in survivorship was statistically significant (χ^2^ = 18.35, df = 2, *p* = 0.0001), possibly resulting from the inadvertently prolonged desiccation times (up to 50 min) in the two first replicates, which shortened to 30 min as soaking methods were streamlined. Removal of the first two replicates from the statistical analysis showed that survivorship did not differ significantly between groups in the last four replicates (74.6 ± 3.7, 77.3 ± 3.7, and 84.375 ± 3.21% for *dsLac2-5*, *dsLac2-3,* and *dsGFP*, respectively; χ^2^ = 3.639, df = 2, *p* = 0.1621). We also compared the survivorship of soaked pupae with those from our microinjection assay. After soaking, the survival of pupae was generally higher and ranged from 56 to 94% in the *dsLac2-5* and *dsGFP* groups. In the microinjection assay, the survival was 50 and 65% for the *dsLac2-5* and *dsGFP* groups, respectively (Figure 3C). The survival in our soaking assay was significantly higher than in the microinjection assay (χ^2^ = 7.7, df = 1, *p* = 0.0055; Appendix A).

In addition to cuticle discoloration, which was first described in *dsLac2 Tr. castaneum*, we observed inadequate and incomplete cuticle hardening in 20–64% of individuals in both the *dsLac2-5* (range = 39–64%) and *dsLac 2-3* (range = 20–45%) groups but not in the dsGFP group (Figure 4A(a–c)). We observed lighter cuticles and delayed darkening in some pupae in the *dsLac2*-treated groups but not in dsGFP-treated groups. The differences in cuticle color could be noted at the 24 h timepoint but not before that (Figure 4B(a–c)). We also noted a significant delay in development (2–4 h) in most mosquitoes in the *dsLac2*-treated groups. At 48 h, nearly all individuals from the *dsGFP* groups emerged normally. At that same timepoint, many of the pupae in the *dsLac2-5* and *dsLac2-3* groups had not yet emerged. In the microinjection assay, we confirmed a *ca.* 6 h delay in emergence in some pupae from the *dsLac2-5* group compared to the *dsGFP* control.

In our assays, we noted similar cuticular defects for both dsRNA constructs we tested, although the numbers of adults with altered phenotypes were significantly higher in the *dsLac2-5*-treated individuals as compared to those in the *dsLac 2-3*′ group (t = 2.964, df = 10, *p* = 0.0142; Figure 4C). In addition to incomplete hardening of the cuticle, we observed other abnormalities (deformed abdominal segments and shorter bodies, curved legs, and wings). The results from our microinjection assay confirmed the phenotypic effects of dsRNA-mediated *laccase 2* knockdown, although the proportion of mosquitoes with visible phenotypic changes further increased to 80% in the *dsLac2-5*-injected group. 

## 4. Discussion

The development of new dsRNA delivery methods is an active area of research [13,36], and here we document the strengths and weaknesses of dsRNA delivery by dehydration and soaking for *Cx. pipiens* pupae. This approach has not been previously used to target this *Cx. pipiens* life stage with RNAi. Our method represents the integration of an improvement over two previously described approaches: one that dehydrated and then soaked *Cx. pipiens* larvae in dsRNA [23], and another that soaked *Ae. aegypti* pupae in dsRNA without dehydration [24]. In our hands, soaking freshly molted *Cx. pipiens* pupae in dsRNA without dehydration failed to elicit a phenotypic response, but dehydration and subsequent rehydration of pupae in concentrated dsRNA solution did. While we attempted multiple dehydration times in preliminary studies, dehydration times <15 min resulted in 5% or fewer individuals with knockdown phenotypes, and longer dehydration times increased mortality. A 30-minute dehydration followed by dsRNA soaking produced low mortality rates while still eliciting a strong phenotypic response. 

Following dehydration and soaking, we observed the clear phenotypic effects of dsRNA exposure in up to 64% of exposed individuals. This effect size is sufficient for the screening of mosquito gene function, and indicates that the method of pupal dehydration and soaking offers an accessible and cost-effective approach to dsRNA delivery in labs without specialized equipment. Once optimized, dehydration had little effect on *Cx. pipiens* pupal survival and most likely enhanced the uptake of dsRNA during the rehydration process. The survival of mosquitoes in our assays ranged from 56 to 75% in the *dsLac2-5′* group, 66 to 83% in the *dsLac2-3′* group, and 80 to 94% in the *dsGFP* group, which is generally higher than in injected pupae [37] (Appendix A). Furthermore, our approach reduced the manipulation and handling of pupae to improve survivorship. As a result, our pupal mortality rates (17–44% in the *dsLac2* groups and 16–20% in the *dsGFP* group) were much lower than documented in previous studies [20]. Our technique eliminated or minimized the risk of damage to pupae and reduced undesirable stress-related effects on mosquito fitness and development [38]. Soaking offers a flexible and rapid screening approach for gene function and would likely allow for the simultaneous screening of multiple genes, enabling a better understanding of the impact of genetic interactions in complex traits.

Challenges for screening of gene function by dehydration and dsRNA soaking do exist, however. As one example, the saline buffer (2 M NaCl) used for dehydration in [23] resulted in significant mortality (over 90%) in fourth instar larvae when used in our assays. We ceased to use it, and instead exposed pupae to desiccation on the benchtop. Prolonged dehydration of pupae (over 40 min) also resulted in increased mortality, and the optimal timeframe between dehydration and rehydration was 30 min. Dehydration success may depend upon species, pupal size, and even relative air humidity, presenting a challenge to the uniform dehydration of pupae and affecting the amount of successfully delivered dsRNA; therefore, this timeframe should be selected and adjusted based on pupae size, cuticle thickness, and tolerance to desiccation.

Timing of dsRNA application for soaking assays was important in our work, as well as in previous studies [20]. We used freshly molted (0–2 h) pupae hours before any signs of cuticle tanning appeared. It had been previously demonstrated that soaking efficacy was reduced when applied to older pupae [20]. We also did not observe any phenotypic changes when 12–20 h old pupae were soaked in the same concentrations of dsRNA used in our assays. This lack of effect might be related to the lower permeability of their cuticles, which could reduce the efficiency of dsRNA uptake. 

When we compared phenotypical defects resulting from dehydration and dsRNA soaking to those from microinjection, we noted similarities in both the *dsLac2-5-* and *dsLac2-3*-treated groups, as well as the *dsLac2-5-*microinjected group. The proportion of mosquitoes with altered phenotypes in our study was variable and significantly higher in the *dsLac2-5* group. This suggests that dsRNA efficacy depends on the design of the dsRNA construct, the specificity of protein synthesis and modifications, and the abovementioned physical properties of pupae. Employing various constructs, as suggested by others [38,39,40,41], or increasing the dsRNA concentration might positively impact dsRNA-mediated gene silencing efficacy in future experiments.

Interestingly, in our microinjection assay, the survival of pupae was similar or higher (up to 65%) compared to the *Culex* spp. pupae microinjections in other studies (up to 40% in [12,13]). In these studies, freshly molted pupae were injected into the base dorsal segments close to the trumpet. In our work, we found that injecting pupae near the antennae by inserting the needle tip between cuticle plates gave higher survivorship, and we encourage the adoption of this pupal injection site in future *Culex* work.

The results from our study also support the critical role for *laccase 2* in the formation of the *Cx. pipiens* pupal cuticle, specifically around the period of intense cuticle darkening. Disruption of the melanin pathway in our assay caused consistent effects similar to those observed in *dsLac2 An. sinensis* mutants (softer and lighter cuticle and delayed emergence of adults). We also observed that, in many mosquitoes from the *dsLac2-5-* and *dsLac2-3*-treated groups, the abdominal segments were shorter, whereas the legs and wings were severely malformed (curved). These individuals also showed delayed development and reduced fitness, and often struggled to leave the pupal enclosure, which resulted in the death of the individual. Observed malformations are likely to be the result of improper cuticle hardening, as was shown for *An. sinensis* [23] and *Tr. castaneum* [22]. These findings suggest that *laccase 2*, via melanin synthesis, might play a substantial role in orchestrating the timely progression of pupal development and facilitating the successful transition from the pupal to the adult stage.

Emerging insecticide resistance remains one of the major challenges in mosquito control [42,43,44,45]. *Laccase 2* and *tyrosine hydroxylase* were included in a list of promising genetic targets for RNAi-based mosquito control to overcome insecticide-resistant populations [5]. Inadequate or incomplete cuticle sclerotization leads to compromised protection from environmental threats, such as UV radiation exposure, viral, bacterial, and fungal pathogens, and decreased fitness. By targeting *laccase 2* and disrupting the melanin synthesis pathway, the survival and general fitness of mosquitoes are severely affected. Interfering with melanin synthesis can lead to compromised immune defenses and increased vulnerability to environmental stresses, ultimately reducing mosquito population sizes. More importantly, *laccase 2* can bind to multiple substrates and is a substantial component in the processes of detoxification in certain insecticides, as has been shown for insecticide-resistant strains of *Ae. aegypti* [46]. Targeting *laccase 2* may disrupt this detoxification mechanism, making mosquitoes more susceptible to insecticides that they were previously resistant to. Ultimately, silencing *laccase 2* expression can be combined with existing mosquito control strategies, such as insecticide-based interventions, to enhance their effectiveness. Overall, our results expand researcher options for target gene functional screening, which can facilitate the development of novel mosquito control applications.

## 5. Conclusions

We show that desiccation and soaking pupae in dsRNA is a promising method for gene functional screening. Soaking freshly molted pupae in dsRNA after desiccation is a simple and generally effective method of screening gene function, producing phenotypic defects in up to 64% of exposed individuals. However, potential limitations to the use of this technique may include variation in desiccation tolerance among the aquatic life stages of other mosquito species and potential stress or damage to pupae during the dehydration process. Variability in the extent of dsRNA uptake also has the potential to hinder the uniformity of gene knockdown among individuals of a mosquito population. While the method shows promise, dsRNA soaking protocols will likely require optimization for each specific target gene and mosquito species being screened. 

## Figures and Tables

**Figure 1 insects-15-00193-f001:**
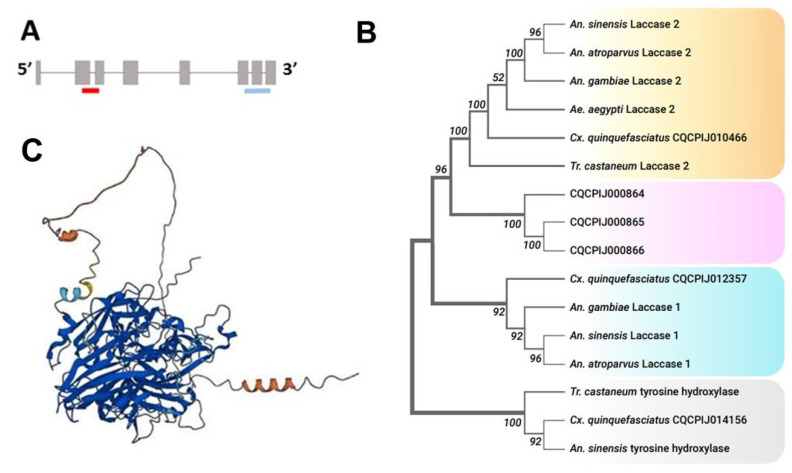
(**A**). Gene structure of *CPIJ010466*. Red and blue colors indicate exon-spanning *dsLaccase 2-5′* and *dsLaccase 2-3′* construct positions. (**B**). Maximum likelihood phylogenetic tree of laccase 1 and laccase 2 proteins among insect species. The numbers above or below the nodes represent bootstrap support values. (**C**). AlphaFold prediction of CPIJ010466 featuring 3 Cu-oxidase domains. AlphaFold structures are colored using a per-residue confidence metric called pLDDT, which is scaled from 1 to 100. Dark blue: very high (pLDDT > 90). Blue: confident (90 > pLDDT > 70). Yellow: low (70 > pLDDT > 50). Orange: very low (pLDDT < 50).

**Figure 2 insects-15-00193-f002:**
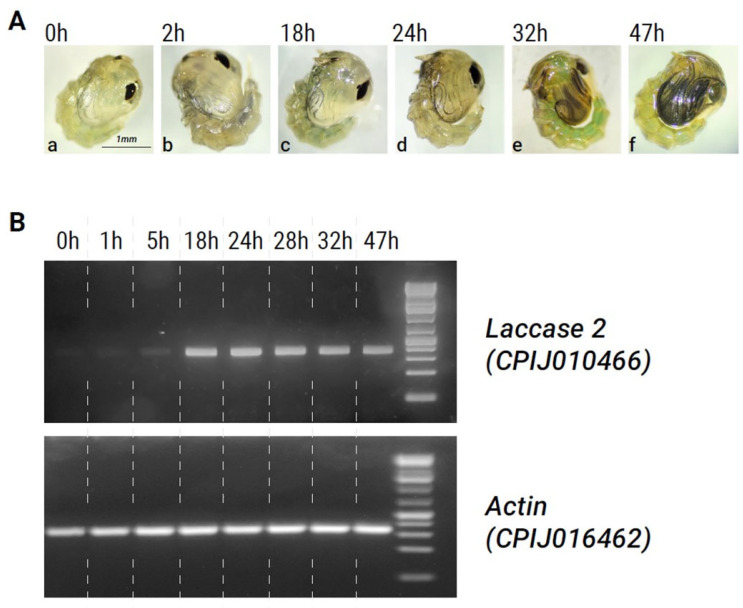
Association between phenotype and *laccase 2* expression during pupal developmental stages. (**A**). (**a**–**f**) Photographs showing melanization of the developing adult cuticle over time in Cx. pipiens pupae. (**B**). Semi-quantitative PCR showing levels of *laccase 2* expression at different stages of development. Actin (CPIJ016462) was used as an internal control. Dashed lines were added to the gel image after it was taken for clarity.

**Figure 3 insects-15-00193-f003:**
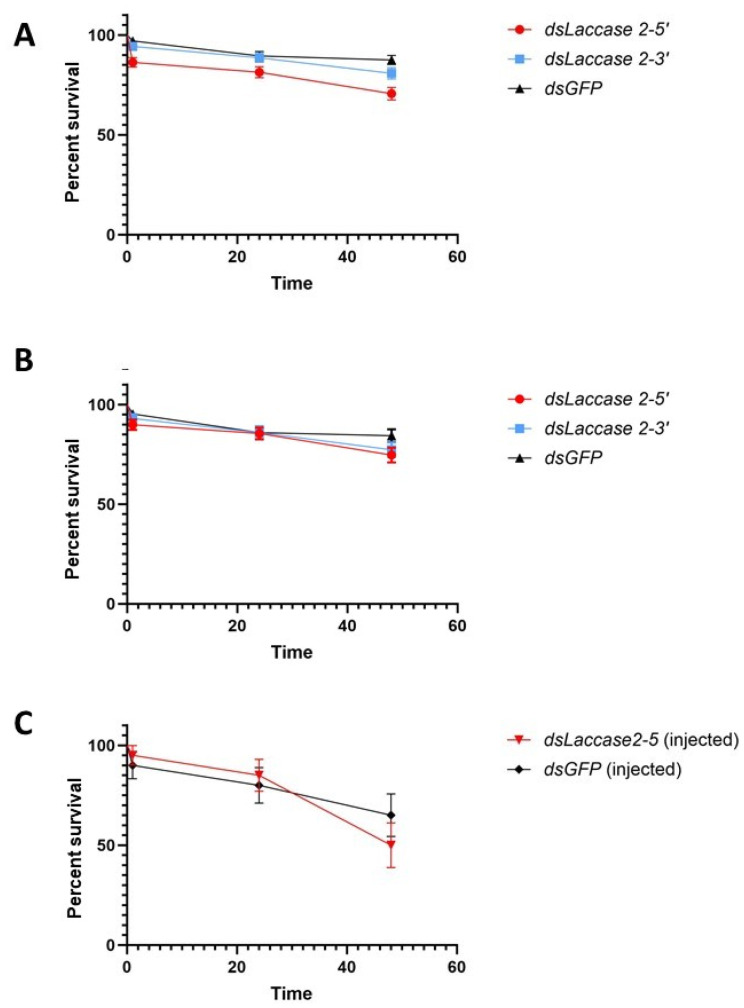
Survival of pupae following soaking. (**A**). Biological replicates 1–6, (**B**). biological replicates 3–6 only, and (**C**). microinjections. Error bars represent ±1 SEM.

**Figure 4 insects-15-00193-f004:**
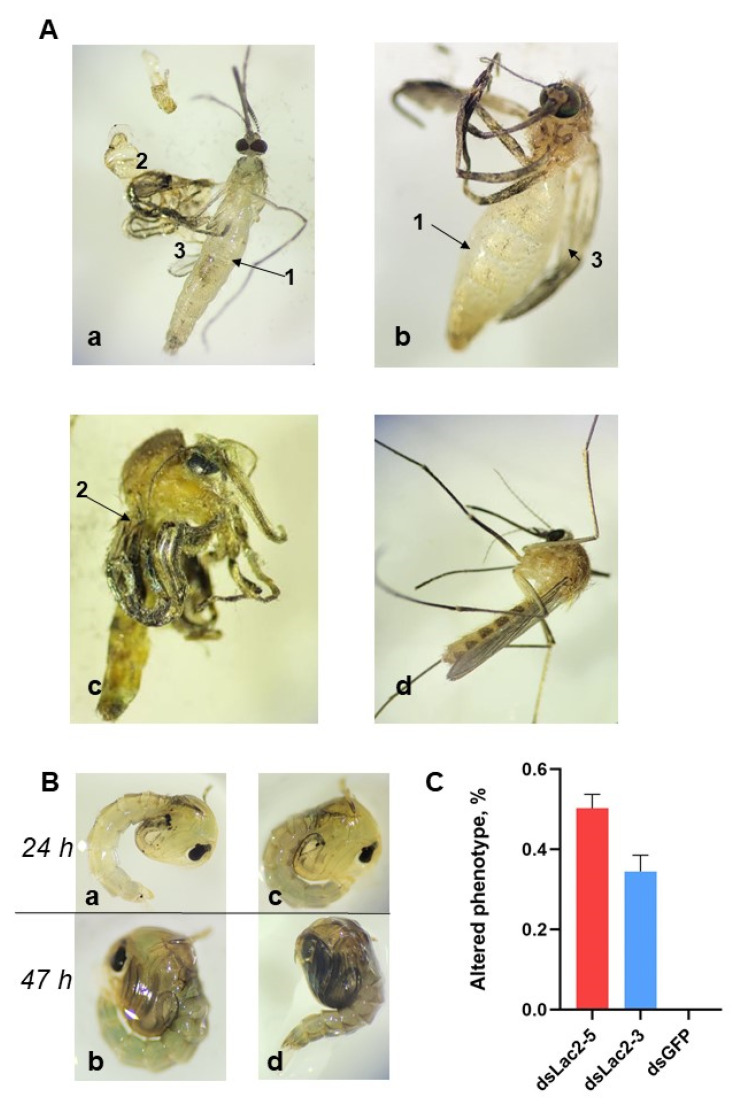
(**A**). Cuticle tanning and cuticle defects of newly emerged adults in (**a**–**c**) *dsLac2* groups compared to (**d**) *dsGFP* control 1 h post-emergence. Arrows indicate changes in phenotype: **1**: transparent cuticle and malformed abdominal segments; **2**: curved legs; **3**: curved malformed wings. (**B**). Cuticle tanning in (**a**,**b**) *dsLac2-* and (**c**,**d**) *dsGFP*-treated pupae at 24 and 47 h post-pupation. (**C**). The percentages of adult mosquitoes with visible phenotypic abnormalities following soaking in *dsLac 2-5* (red) and *dsLac 2-3* (blue), as compared to *dsGFP* controls. Data are expressed as a mean percentage across replicates ±1 SEM.

**Table 1 insects-15-00193-t001:** Oligonucleotides designed for this study.

Reaction Type	Gene Target	Forward Primer (5′---3′)	Reverse Primer (5′---3′)
Semi-quantitative PCR	*CPIJ010466*	ACTCGCACATCTCCGACTCT	CCAGCACGGTGTAGTACTCG
qRT-PCR	*CqLaccase 2*	GTTCCCGTAACGAGCACACATT	CTCCAAACCGTCTTCGCATC
	*CqActin*	TGCGTGACATCAAGGAGAAG	GTGTTGGCGTACAGGTCCTT
dsRNA template amplification	*CqLaccase 2*–*5′*	GGATCCTAATACGACTCACTATAGGTCCGTCATCTGGACTTTAGC	GGATCCTAATACGACTCACTATAGGAGTATTTCCCTGCTGGATGG
	*CqLaccase 2*–*3′*	TAATACGACTCACTATAGGACGTAATCTCGCTGATCGAC	TAATACGACTCACTATAGGACAGCGATCGTGTCCTTTAG
	*GFP*	TAATACGACTCACTATAGGCTCGGTGTTGCTGTGATCCTC	TAATACGACTCACTATAGGGAGAATGGAGAGCGACGAGAGC

**Table 2 insects-15-00193-t002:** Relative quantification of *laccase 2* expression based on fold-change values in semi-quantitative PCR gel band intensities for pupae collected at 0, 1, 5, 18, 24, 28, 32, and 47 h.

Sample Timepoint (h)	Fold Change in Band Intensity
0	0.06
1	0.08
5	0.13
18	1
24	1
28	0.97
32	0.83
47	0.81

**Table 3 insects-15-00193-t003:** Differential expression of *laccase 2* in *dsLac2-5-* and *dsLac2-3*-treated pupae (combined for soaking, *dsLac2*) as compared to *dsGFP* controls from n biological replicates (pools of 3 pupae collected at 1, 24, or 47 h post-treatment). * denotes statistical significance at a priori α = 0.05 level.

dsRNA Delivery	Sample Timepoint (h)	Treatment	n	Mean Fold Change	Standard Error	*p*-Value
Soaking	1	*dsLac2*	6	0.56	0.06	--
		*dsGFP*	3	1.18	0.51	0.273
	24	*dsLac2*	6	0.49	0.11	--
		*dsGFP*	3	1.01	0.09	0.017 *
	47	*dsLac2*	6	0.69	0.29	--
		*dsGFP*	3	1.09	0.3	0.185
Injection	24	*dsLac2-5*	1	0.21	--	--
		*dsGFP*	1	1	--	--
	47	*dsLac2-5*	1	0.18	--	--
		*dsGFP*	1	1	--	--

## Data Availability

All data are provided in the manuscript.

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
