# Peer review of "RNAi-Mediated Silencing of Laccase 2 in Culex pipiens Pupae via Dehydration and Soaking Results in Multiple Defects in Cuticular Development"

_insects, 2024, doi:10.3390/insects15030193_

Round 1
Reviewer 1 Report
Comments and Suggestions for Authors
Successful knockdown of a gene using RNA interference in mosquito larvae and pupae remains a challenge because of their aquatic habitats. The authors in this study attempted to solve this problem by taking advantage of rehydration-driven soaking of mosquito pupae in concentrated dsRNA solution. This has allowed the authors to obtain a moderate success in silencing the gene laccase 2 that controls the hardening of the cuticle.
The study was well designed and executed, and the results are convincing, except for a few formatting issues.
A pertinent question is related to the optimal dehydration duration for pupae in gene silencing experiments. It is worth investigating if a period shorter than 30 min could maintain effective gene silencing while potentially improving survivorship.
Comments on the Quality of English LanguageEnglish writing is fine.
Author Response
Please see the attachement.

Reviewer 2 Report
Comments and Suggestions for Authors
Comments to authors:
In the manuscript titled “RNAi-mediated silencing of laccase 2 in Culex pipiens pupae via dehydration and soaking results in multiple defects in cuticular development”, the authors investigated the efficacy of the dehydration and soaking method for delivering dsRNA to silence the laccase 2 gene in Culex pipiens pupae. Previous studies demonstrate that sustained target gene silencing can be achieved in Culex pipiens larva, pupal, and adult stages through soaking, despite the soaking occurring during larval stages. Additionally, I have observed some issues in the manuscript, which I list below.
1. Line No: 115. It is inaccurate to claim high throughput as the study only demonstrated effects for one gene. Additionally, there is less knockdown of the target gene at 48 h compared to 24 h post-soaking, and lower mortality after knockdown compared to other studies targeting the same gene.
2. Lines No: 117-178: Performing 35 cycles for semi-quantitative PCR may lead to plateauing of the amplicon, hindering accurate quantification of exponentially amplifying amplicons. Ideally, It is recommended to stop PCR cycles during the exponential phase for accurate semi-quantitation. I also suggest the authors verify differences in transcript levels at various developmental time points by quantifying band intensities in the gel.
3. I am curious why the authors opted for semi-quantitative PCR instead of qPCR, considering the latter provides superior quantification, especially when they have access to it.
4. Line No. 249: Please provide a clearer description of how microinjection was performed.
5. Table 2: It is better to have at least 3 biological replicates in case of microinjection to say the observations are indeed true not by chance. Therefore, I recommend assessing the knockdown of the target gene in additional biological replicates following microinjection.
6. Fig. 3. Please club all biological replicates into a single line chart and compare them with microinjection survival data to facilitate easier understanding.
7. Did the authors observe any discoloration of pupae and reduced sclerotization of pupal cuticle after soaking newly emerged pupae? This information is crucial for understanding the timeline of this method effects. Please include photographs depicting any differences in cuticle color between laccase 2 dsRNA treatments and the control group.
Round 2
Reviewer 2 Report
Comments and Suggestions for Authors
After reviewing the revised manuscript, I noted that the authors have made some improvements and also explained their limitations regarding biological replicates in the case of microinjection. In light of these improvements, I believe the manuscript is now suitable for publication.